# Short-Term Head-Out Whole-Body Cold-Water Immersion Facilitates Positive Affect and Increases Interaction between Large-Scale Brain Networks

**DOI:** 10.3390/biology12020211

**Published:** 2023-01-29

**Authors:** Ala Yankouskaya, Ruth Williamson, Cameron Stacey, John James Totman, Heather Massey

**Affiliations:** 1Department of Psychology, Faculty of Science and Technology, Bournemouth University, Poole BH12 5BB, UK; 2University Hospital Dorset, NHS Foundation Trust, Bournemouth BH15 2JB, UK; 3Department Radiography and Medical Imaging, Fatima College of Health Science, Abu Dhabi P.O. Box 3798, United Arab Emirates; 4Extreme Environments Laboratory, School of Sport Health and Exercise Science, Faculty of Science and Heath, University of Portsmouth, Portsmouth PO1 2UP, UK

**Keywords:** cold-water immersion, large-scale brain networks, affects

## Abstract

**Simple Summary:**

An increasing number of people are turning to cold showers or outdoor swimming to keep fit and for health benefits. After cold-water immersion, the swimmers reported elevated positive emotions and a decreased negative emotional state. The present study aimed to examine how mood changes after cold-water immersion are associated with changes in brain connectivity. Thirty-three healthy adults naïve to cold-water swimming (age range 20–45, 16 females) undertook a 20 °C 5 min whole-body bath. We measured brain connectivity and self-reported emotional state before and after cold-water immersion. Our findings showed that participants felt more active, alert, attentive, proud, and inspired and less distressed and nervous after having a cold-water bath. The changes in positive emotions were associated with the coupling between brain areas involved in attention control, emotion, and self-regulation. A reduction in negative emotions did not show strong associations with changes in brain connectivity. The results indicate that short-term whole-body cold-water immersion may have integrative effects on brain functioning, contributing to the reported improvement in mood.

**Abstract:**

An emerging body of evidence indicates that short-term immersion in cold water facilitates positive affect and reduces negative affect. However, the neural mechanisms underlying these effects remain largely unknown. For the first time, we employed functional magnetic resonance imaging (fMRI) to identify topological clusters of networks coupled with behavioural changes in positive and negative affect after a 5 min cold-water immersion. Perceived changes in positive affect were associated with feeling more active, alert, attentive, proud, and inspired, whilst changes in negative affect reflected reductions in distress and nervousness. The increase in positive affect was supported by a unique component of interacting networks, including the medial prefrontal node of the default mode network, a posterior parietal node of the frontoparietal network, and anterior cingulate and rostral prefrontal parts of the salience network and visual lateral network. This component emerged as a result of a focal effect confined to few connections. Changes in negative affect were associated with a distributed component of interacting networks at a reduced threshold. Affective changes after cold-water immersion occurred independently, supporting the bivalence model of affective processing. Interactions between large-scale networks linked to positive affect indicated the integrative effects of cold-water immersion on brain functioning.

## 1. Introduction

The popularity of cold-water immersion (CWI), swimming outdoors, or cold showering has grown in recent years [1]. With much evidence suggesting the activity has risks [2,3], emerging research indicates the potential for improvements in health and well-being [4]. Research investigating the effects of cold-water immersion on physiological responses suggested that even a short exposure to cold water triggers a cascade of adaptative biochemical and physiological reactions [5] that have beneficial effects on the immune system [6], hemodynamic [7], and motor functions [8].

An emerging body of evidence indicates that regular swimming in cold water reduces fatigue, lessens depressive symptoms [9,10], and improves general well-being [11]. Moreover, some studies reported the immediate effects of CWI on elevating mood and increasing a positive emotional state. For example, participants enrolled in a 10-week open-water programme showed a significant increase in the positive profile of mood state and a decrease in the negative mood, with total mood disturbance reduced following each swim [12]. The therapeutic benefits of cold water may be gained even from a single exposure. In the study by Kelly and Bird [13], participants were immersed in chest-deep open water. Following a single immersion, participants reported less negative mood disturbance, and showed significantly increased vigour and esteem-related effects with effect sizes ranging from medium to large.

At the biochemical level, whole-body exposure to cold triggers a release of neurotransmitters such as serotonin, cortisol, dopamine, norepinephrine, and β-endorphin [14], which play a crucial role in emotion regulation [15,16], stress regulation [17], and reward processing [18]. Deficits in these neurotransmitters have been reported as critical factors in developing psychiatric disorders such as depression, anxiety, and emotional disturbances [19].

Although the biochemical changes following cold-water immersion help shed light on the immediate mood-lifting, the neural mechanisms underlying these effects remain largely unknown. Only a few studies focusing on how the effects of cold-water immersion change brain activity have been conducted in humans. However, their objectives were to test the effects of experimentally administered tonic limb cooling on the neural mechanisms of the negative affect induced by low temperature [20,21,22]. It must be noted that physiological responses to limb cooling using very low water temperature (1–10 °C) constitute points of contrast to painful experiences that differ from what we experience after a cold shower or swimming in open water. As such, we still do not know how a whole-body cold-water immersion affects neural processes relating to affect. We put forward this question as the main objective of the present study.

Positive affect has been defined as the extent to which a person feels enthusiastic, active, and alert. High positive affect reflects high energy, concentration, and pleasurable engagement. In contrast, negative affect has been characterised as an individual’s experience of distress and negative response to the environment, including nervousness, fear, anger, and guilt [23]. A long debate about the structure of affective states inspired three psychological models to explain the relationship between positivity and negativity [24]. Two models (the ‘bivalence hypothesis’ [25] and the ‘bipolar hypothesis’ [26]) argued for the relative independence of positive and negative affect. They proposed that two separate physical systems are involved in generating positive and negative affect. However, these hypotheses made different predictions about how these affects are represented in the brain [24]. According to the ‘bipolar hypothesis’, brain regions that showed increased neural responses to positive affect would also show decreasing responses to negative affect and vice versa. Support for this hypothesis has been well-documented in studies reporting an increasing amygdala’s activity in responding to unpleasant stimuli and decreasing in responding to pleasant pictures [27]. In contrast, the ‘bivalence hypothesis’ assumed that positive and negative affect were processed by spatially separate sets of brain areas which might be reciprocally activated/deactivated. For example, multiple studies suggested left lateralisation in the medial prefrontal cortex during positive stimulus processing.

In contrast, negative stimulus processing was lateralised towards the right in the dorsolateral prefrontal cortex and towards the left in the amygdala and uncus [28]. The third model (so-called ‘flexible affective workspace’) proposed that, regardless of the valence, affect is processed by a flexible and spatially distributed set of brain regions (i.e., no single region uniquely represents positivity or negativity) [28,29]. A meta-analysis of ‘activation’ neuroimaging studies supported this model by demonstrating that regions such as the medial prefrontal cortex (MPFC), anterior cingulate cortex (ACC), insula, amygdala, ventral striatum, thalamus, and occipitotemporal cortex can form interconnected networks to process positive and negative affect [24,30].

Although none of these models has received decisive empirical evidence, all of them can provide plausible hypotheses for the effects of cold-water immersion on changes in affective state. Suppose the mood-lifting effect from whole-body immersion in cold water is triggered by simultaneous changes in positive and negative affect in the opposite directions. In that case, we should see a correlation between neural responses and behavioural changes. Observing this correlation in specific brain areas would support the bipolar hypothesis indicating the mutual relationship between positive and negative affect at the regional level. Alternatively, some brain areas may be preferentially involved in processing positive or negative affect. In this case, we should be able to identify spatially distinct areas or networks correlated with changes in positive or negative states after cold-water immersion, proving support for the bivalence hypothesis. Rejecting these two hypotheses may provide evidence in favour of the flexible affective workspace hypothesis; specifically, if the regions involved in the affective workspace will show a valence-independent activity. In the present study, we aimed to test these hypotheses using behavioural and neuroimaging data collected before and after whole-body cold-water immersion in a group of unhabituated people.

Our methodological approach stems from a recently developed framework, which characterises the spatiotemporal organisation of the brain into large-scale networks based on patterns of synchronous brain activity [31,32]. The patterns representing statistical dependencies in the synchronised fluctuations in functional MRI signals were established using resting-state connectivity under task-free conditions. It was demonstrated that these statistical dependencies were presented regardless of cognitive context, much like structural connectivity [33]. The intrinsic network architecture was not fixed, however, and could be modified as necessary to implement the processing of environmental stimulation or task demands resulting in changes in functional connectivity within the system (e.g., a partial breakdown of network communities) or between large-scale networks [34]. The large-scale networks identified by clustering patterns of resting-state connectivity showed sensitivity to the individual measures of positive and negative affect [35]. Moreover, aberrant connectivity between large-scale networks (e.g., between the default mode network, salience network, and frontoparietal network) has been reported in many mood disorder conditions [36,37]. Here, we used the large-scale network approach to identify network communities coupled with self-reported behavioural changes in positive and negative affect after cold-water immersion.

## 2. Methods

### 2.1. Ethics Statement

The study was approved by the ethics committees of Bournemouth University (Ethics ID 34976) and conducted in accordance with the Declaration of Helsinki. Written informed consent from each participant was obtained before the study.

### 2.2. Participants

Thirty-nine participants were recruited through advertisements distributed on campus at Bournemouth University, University Hospital Dorset, University of Portsmouth, and social media platforms (Facebook, Twitter). Responders received detailed information about eligibility criteria and experimental procedures in the participant information sheet. The eligibility criteria were: (i) free from chronic pain of any type; (ii) no medication use; (iii) no history of medical, neurological, psychiatric, or substance use disorders; (iv) not pregnant (for females); (v) naïve to cold-water immersion in the last 12–18 months. Four responders were excluded due to ongoing mental health or chronic physical conditions. One participant was excluded based on MRI safety pre-screening issues, and one was due to habituation to cold-water immersion. The remaining thirty-three participants (right-handed, age range 20–45 years old, M = 28.36, SD = 8.74, 16 females) met the inclusion criteria. They were scheduled to visit the Institute of Medical Imaging and Visualisation (IMIV) at Bournemouth University, where the study occurred.

### 2.3. Experimental Design and Procedures

The experimental design of the present study is depicted in Figure 1.

At arrival, participants were shown the experimental facilities and received a detailed explanation of the course, procedures, and study duration. In addition, participants were asked about their previous experience undertaking MRI/fMRI procedures. After providing written consent for the study, participants completed an MRI safety assessment and the PANAS and were taken to the pre-CWI scanning session. The pre-CWI scanning session included a 4.5 min anatomical scan and a 13.12 min resting state acquisition, where participants were instructed to rest with their eyes open. We chose the eyes-open instruction based on previous evidence that the amplitude of the resting-state BOLD signal was decreased in the eyes-open condition compared to the eyes-closed condition [38,39]. Immediately following the pre-CWI, participants exited the scanner and were asked to change into their swimwear.

After the change, participants were asked to rest in a seated position whilst ECG electrodes were attached, and a mouthpiece positioned. Two minutes of baseline measures were recorded in a seated posture before stepping into the cold bath. After the baseline testing, participants were immersed to the level of their clavicles (water temperature at 19.93 °C ± 0.13 °C). They were closely monitored during the immersion and remained in the cold bath for 5 min unless they requested to exit, or the physiologist requested an early finish. The early exit was related to 6 or more ectopic beats appearing on the ECG trace in a minute, extreme pain on entry, or feeling unwell for any reason. One immersion was stopped early due to many ectopic beats; they were monitored post-immersion and were happy to undertake the second MRI scan.

Upon completion of the 5 min immersion, participants exited the water, were de-instrumented, dried and dressed quickly, and returned to the MRI scanner. The duration from the immersion completion to the scan’s start was recorded. The final resting state scan of 13.12 min was completed, followed by the post-immersion PANAS questionnaire. At the end of the experiment, participants received a GBP 20 Amazon voucher as compensation for their time.

### 2.4. Instruments and Measurements

#### 2.4.1. Positive and Negative Affect Schedule (PANAS)

The Positive and Negative Affect Schedule (PANAS) is a 20-item self-report measure of positive (PA), and negative (NA) affect [23]. The term ‘positive affect’ refers to experiencing a positive emotion that ranges on the orthogonal dimension of arousal and represents the extent to which an individual experiences pleasurable engagement with the environment. Emotions such as enthusiasm and alertness indicate high PA, whilst lethargy and sadness characterise low PA. In contrast, a high NA reflects subjective distress and unpleasurable engagement (e.g., involving a variety of aversive mood states, such as anger, guilt, and fear), and the absence of these feelings characterises low NA.

The PANAS demonstrated good psychometric properties with internal consistency ranging between 0.86 and 0.90 for positive affect and 0.84 and 0.87 for negative affect [23]. Scores ranged from 10 to 50 for both sets of items. For the total positive score, a higher score indicated more of a positive affect. For the total negative score, a lower score indicated less of an adverse affect. The schedule was administered twice during the study (immediately pre-CWI and post-CWI), asking participants to evaluate their mood ‘right now’ using a 5-points scale (1—very slightly or not at all, 2—a little, 3—moderately, 4—quite a bit, 5—extremely).

#### 2.4.2. Controlling Physiological Responses to Cold-Water Immersion

The water temperature was controlled by a combination of a heater, pump (Laz-y-spa, Leeds, UK), and chiller unit (Grant Instruments, Royston, UK). pH and chlorine levels were monitored twice a day, and the tank was back-washed daily.

The following monitoring was used to monitor the physiological response to immersion. Participants dressed in swimwear were instrumented with a three-lead electrocardiogram (ECG, Fukuda Denshi, Sheffield, UK). The electrodes were covered with Tegaderm dressings (3M, Neuss Hammfeld, Germany) to ensure adhesion and reduce signal artefacts. To measure inspiratory volumes and respiratory frequency, participants were fitted with a nose clip and a mouthpiece attached to a one-way valve (Hans Rudolf, Shawnee, KS, USA) and turbine (K.L. Engineering, Fitchburg, WI, USA). The ECG and respiratory equipment were connected to an analogue digital recorder (Powerlab, AD Instruments, Bella Vista, NSW, Australia) and recorded onto a laptop at a sampling rate of 400 Hz (Chart software, AD Instruments Australia).

Controlling for the physiological responses ensured that participants had the expected initial response to cold-water immersion (i.e., an ‘inspiratory gasp’, hyperventilation, and tachycardia). The hyperventilation seen after the initial gasp of the cold shock response reflects a stress response, accompanied by a sympathetic overdrive peaking within 30 s and adapted over 3–5 min of immersion in most individuals. However, the residual biochemical and physiological effects of sympathetic stress response last between 20 and 30 min before the para-sympathetic nervous system takes over and returns the body to a resting state [40,41].

#### 2.4.3. MRI Acquisition

Image acquisition was performed on a 3-Tesla Siemens Magnetom Lumina scanner (Siemens Healthcare, Erlangen, Germany). We obtained T1-weighted MPRAGE images with the following parameters: repetition time (TR) = 1900 ms, echo time (TE) = 2.74 ms, flip angle 8°, field of view (FOV): 256 × 256 mm, voxel size: 1.0 × 1.0 × 1.0 mm, and 192 axial slices.

The blood oxygenation level-dependent (BOLD) contrast whole-brain functional images were acquired using a T2-weighted gradient-echo Echo Planar Imaging (EPI) sequence with a 32-channel head coil. The sequence duration was 13.12 min. The acquisition parameters were TR = 2680 ms; TE = 30 ms; matrix size = 64 × 64; slice thickness 2.5 mm; in-plane resolution 3 × 3 mm; and flip angle = 80°. Two-hundred-and-ninety-two volumes with 48 axial slices were measured in interleaved slice order and positioned along a line to the anterior-posterior commissure (AC-PC orientation). An automated high-order shimming technique was used to maximise magnetic field homogeneity.

### 2.5. Data Analysis

#### 2.5.1. PANAS

Individual PA and NA scores were calculated for pre- and post-CWI assessment and entered into a linear mixed-effect regression model (LMM) [39]. We estimated a full factorial model entering affect (positive, negative), condition (pre-, post-CWI), and interaction between affect and condition as fixed-effect independent variables. We also added the subject into the model as a random effect (random intercept model), allowing the model to estimate a separate intercept for each participant and the effects of other variables relative to each individual’s mean response. The contribution of the random effect of subjects was estimated using the likelihood ratio test (LRT). The likelihood ratio statistic is equal to two times the difference of the log-likelihoods of two models, where one model includes a parameter of interest (fitted model), and the second model (null model) does not contain the parameter of interest.

#### 2.5.2. Measuring Physiological Responses of Cold-water Immersion

Water temperature was measured throughout the baseline rest and immersion at a sampling frequency of 0.1 Hz. For each participant, the heart rate (beats.min^−1^) derived from ECG data, respiratory frequency (breaths.min^−1^), and tidal volume (L) calculated from the turbine were recorded at 10 s intervals from the start of the baseline rest period to the end of the cold-water immersion. Inspiratory gasp (L) was recorded as the largest inhalation within the first 10 s of the foot entering the water. Mean inspiratory gasp was calculated, and mean heart rate, tidal volume, and respiratory frequency were calculated at 10 s intervals. The duration of time between exiting the cold water and starting the scan was recorded.

#### 2.5.3. fMRI Data Pre-Processing

We used the default pre-processing pipeline for volume-based analyses implemented in CONN (functional realignment and unwarp, slice-timing correction, outlier identification, direct segmentation and normalisation, and functional smoothing) [42]. Acquisitions with framewise displacement above 0.9 mm or changes in the Global Blood-Oxygen-Level-Dependent (BOLD) signal above 5 s.d. were flagged as potential outliers (43). The gross head movements’ detection results indicated that our sample did not contain participants with a head displacement exceeding 0.9 mm in more than 2.5% of volumes in the pre- and post- cold-water immersion sessions. To ensure that head movement artefacts did not contaminate our findings, functional data with frame-to-frame displacements greater than 0.9 mm were censored after bandpass filtering [43]. Subject-level summary measurements of motion displacement were used as covariates in the second-level analysis. After the outlier removing step, we performed a quality control (QC)–functional connectivity (FC) check implemented in CONN to assess the residual effects of subject motion. This method computes functional connectivity between randomly selected pairs of points within the brain and evaluates whether these connectivity values are correlated with other QC measures, such as subject–motion parameters. The QC–FC showed that the QC–FC perfectly matched the null hypothesis, indicating that functional connectivity did not associate with the residual effects in the dataset.

Functional and anatomical data were co-registered; segmented into grey matter, white matter, and CSF tissue; and normalised into standard MNI space using a unified segmentation and normalisation procedure [44]. After that, functional and anatomical data were resampled to a default 180 × 216 × 180 bounding box with a voxel size of 2 mm^3^ for functional data and 1 mm^3^ for anatomical data. Functional images were then smoothed with an 8 mm isotropic Gaussian kernel to increase the BOLD signal-to-noise ratio and reduce the influence of residual variability in functional and gyral anatomy across subjects [45].

#### 2.5.4. fMRI Data Denoising

Removing physiological noise from respiration and cardiac pulsation that also aliases to the lower frequency band is critical for resting-state connectivity analyses [46]. We used CONN’s default denoising pipeline that combines the component-based noise correction technique (CompCor), estimation of subject–motion parameters, scrubbing, and temporal band-pass filtering. With CompCor, noise ROIs were defined within the white matter and CSF masks individually for each participant. Then, the signal from the noise ROIs was decomposed with a principal component analysis (PCA), and the resulting components’ time courses were regressed from the data. Because CompCor addresses potential confounding effects without the risk of artificially introducing anticorrelations into the functional connectivity estimates, the global signal regression was not applied to the data. A total of 12 potential noise components were defined from the estimated subject–motion parameters to minimize motion-related BOLD variability. Identified outliers were removed to limit their influence on the BOLD signal. Finally, temporal frequencies below or above the range of 0.09–0.008 Hz were removed from the BOLD signal to focus on slow-frequency fluctuations while minimizing the influence of physiological, head-motion, and other noise sources.

Network connectivity analysis. After the pre-processing and denoising steps, we used ROI-to-ROI connectivity (RRC) measures of large-scale networks. The large-scale networks’ ROIs were defined from the default CONN’s networks’ atlas (derived from ICA analyses based on the Human Connectome Project (HCP) dataset of 497 subjects). The networks’ atlas delineates an extended set of classical networks: default mode network (4 ROIs), sensorimotor (2 ROIs), visual (4 ROIs), salience/cingulo-opercular (7 ROIs), dorsal attention (4 ROIs), frontoparietal/central executive (4 ROIs), language (4 ROIs), and cerebellar (2 ROIs). A detailed list of these networks and their notes is provided in Appendix A. The cerebellar ROIs were not included as they only had partial coverage in the participants. In total, we analysed 870 connections across 30 ROIs.

To define the network components associated with two conditions (pre- and post-CWI), we used network-based statistics (NBS) [47]. First, we defined condition-specific (pre-CWI, post-CWI) functional connectivity strength by computing weighted RRC matrices using a weighted least squares linear model with temporal weights for each condition/participant. The weights were defined to encompass the entire scanning session. Weighted RRC matrices of Fisher-transformed bivariate correlation coefficients between all ROIs/nodes (30 × 30) were calculated for each condition and participant. These matrices were submitted to the second-level analysis, where the differences between pre-CWI and post-CWI conditions [contrast post-CWI > pre-CWI] were calculated for every edge/connection using a general linear model (GLM). We also calculated the changes in PANAS-positive and negative affect for each participant such as [positive post-CWI − positive pre-CWI], [negative post-CWI − negative pre-CWI]. These vectors were added as covariates for the second-level analyses.

The resulting statistical parametric map for each contrast was thresholded using a priory connection threshold (‘height’ threshold) (uncorrected *p* < 0.001) to construct a set of suprathreshold links among all ROIs/nodes of the between-condition differences. The connection threshold is a user-determined parameter in an NBS analysis. For example, effects presented at only a conservative connection threshold (e.g., *p* < 0.001) are likely to be characterised by strong, topologically focal differences between conditions constituting the effect. Effects presented only at a relatively liberal threshold (e.g., *p* < 0.05) are likely to be subtle yet topologically extended. Effects presented at both thresholds combine features of topologically focal and distributed differences [47]. Although our analysis focused on the former threshold, we also explored changes in connectivity using the lower threshold to reveal more information about the nature of the effect.

Next, we identified any connected components (topological clusters) in the set of suprathreshold links. We defined the size of each component as the sum of T-squared statistics over all connections within each component. The critical assumption inherent to the NBS here is that connections for which the null hypothesis is false are arranged in an interconnected configuration rather than being confined to a single connection or distributed over several connections that are in isolation. Therefore, the presence of a component may be evidence of a non-chance structure for which the null hypothesis can be rejected at the level of the structure as a whole, but not for any individual connection alone [48]. Finally, an FWE-corrected *p*-value for each component was computed using permutation testing. The basic assumption of the permutation procedure is that under the null hypothesis, random rearranging correspondence between data points and their labels does not affect the test statistics. This would not be the case if the null hypothesis were false. The labels for the tested contrast were randomly rearranged for corresponding data points according to a permutation vector of integers from 1 to the total number of data points. The same permutation vector was used for every connection (870 in total) to preserve interdependencies between connections and to remain within the same participant. The size of the largest component was recorded for each permutation, yielding an empirical null distribution for the largest component size. This procedure was performed 1000 times. The FWE-corrected *p*-value for a component of a given size was then estimated as the proportion of permutations for which the largest component was of the same size or greater, thus, representing the likelihood under the null hypothesis of finding one or more components with this or a larger mass across the entire set of networks.

To characterise the properties of each component, we reported ‘size’ as the number of suprathreshold connections, ‘intensity’ (mass) measures as their overall strength (i.e., the sum of absolute T-values over these suprathreshold connections), and *p*-values associated with these measures. In addition, we provided complementary statistics for each connection, such as the effect size for significant components and the between-subject variability for each connection within a component, to gain more insight into the effect of interest. It has to be noted that we did not violate the NBS inference about a component as a whole by providing effect sizes for each connection. This information about effect sizes helped to interpret the nature of connections within each component.

## 3. Results

### 3.1. Increased Cardio-Respiratory Activity Resulting from CWI

The heart rate increased significantly (t(33) = 4.83, *p* < 0.001, MD = 10.61, Cohen’s d = 0.84, 95%CI [0.44, 1.23], BF_10_ > 100) from baseline to 30 s of CWI. Similarly, the breathing tidal volume increased significantly from resting baseline levels (t(30) = 6.41, *p* < 0.001, MD = 0.44, Cohen’s d = 1.15, 95%CI [0.69, 1.60], BF_10_ > 100) and remained elevated for the time of CWI (Appendix A).

### 3.2. Self-Report of Increasing Positive and Decreasing Negative Affect after CWI

An LMM on positive and negative scores (affect) for pre-and post-CWI (conditions) showed a main effect of affect (F(1,93) = 368.35, *p* < 0.001) and interaction between affect and conditions (F(1,93) = 37.12, *p* < 0.001). The main effect of conditions was non-significant (F(1,93) = 1.50, *p* = 0.22). Post hoc tests indicated that the interaction between affect and conditions was driven by the increasing positive affect for post-CWI compared to pre-CWI assessment (MD = 7, SE = 1.35, *t*(93) = 5.17, *p* < 0.001), and decreased in the scores for negative affect for post-CWI compared to pre-CWI (MD = −4.66, SE = 1.35, *t*(93) = −3.44, *p* = 0.005) (Figure 2). A random effect of subjects (*LRT* = 1.33, *df* = 1, *p* = 0.24) did not contribute significantly to the model (9% to the overall variation in the PANAS scores).

An analysis of the individual items of PANAS using Wilcoxon’s signed rank test and Bayesian analysis showed decisive (BF_10_ > 100) and very strong evidence (BF_10_ = 30–100) for feeling more active, alert, attentive, inspired, proud, and less nervous after cold-water immersion (Figure 3).

To test whether the self-reported increase in positive affect was associated with the decreasing negative affect, we performed a Pearson correlation on the pre/post-negative/positive affect scores. The correlation analysis showed associations between pre- and post-CWI for the positive affect (r = 0.48, *p* = 0.005, 95% CI [0.26, 0.69], BF_10_ = 9.54) and negative affect (r = 0.45, *p* = 0.01, 95% CI [0.15, 0.72], BF_10_ = 5.02). No association was found between the positive and negative affect in pre-CWI (r = −0.27, *p* = 0.13, BF_10_ = 0.66) or post-CWI (r = −0.05, *p* = 0.79, BF_10_ = 0.23). There was also no evidence for an association between changes in positive affect (defined as [post-CWI positive − pre-CWI positive]) and changes in negative affect (defined as [post-CWI negative − pre-CWI negative]) (r = −0.28, *p* = 0.11, 95% CI [−0.58, 0.07], BF_10_ = 0.74).

In the present study, 14 participants reported previous experience in MRI/fMRI, and 19 were naïve to the scanning procedures. To examine whether familiarity with the procedures affected self-reported pre- and post-CWI PANAS scores, we performed between-group comparisons (Group 1 experienced, Group 2 naïve) using Welch’s *t*-test. There were no significant differences between these groups in self-reported scores for the pre-CWI positive affects (t(27.8) = −1.01, *p* = 0.32, MD = −2.60, 95% CI [−7.90, 2.70]) and negative affects (t(23.5) = 0.14, *p* = 0.89, MD = 0.24, 95% CI [−3.37, 3.86]). No differences between these groups were found for the post-CWI affect (t(31.0) = 0.003, *p*= 0.99; t(22.6) = 0.18, *p* = 0.86, MD = 0.19, 95% CI [−2.02, 2.41] (see details in Appendix A).

### 3.3. Changes in Functional Connections in Post-CWI Compared to Pre-CWI

Data from two participants were partially lost due to technical failure. The remaining 31 complete datasets were used for the analyses.

To examine the effect of cold-water immersion on functional connectivity between large-scale networks, we tested the contrast [post-CWI > pre-CWI] using the connectivity ‘height’ threshold *p* < 0.001. Two clusters survived a network-level p-FWE corrected cluster threshold (*p* < 0.05).

One cluster (mass = 165, size = 4, p-FWE < 0.001) comprised positive coupling between the medial prefrontal cortex of the default mode network (DMN.MPFC) and left anterior insula of the salience network (SN.AIns L) (T(30) = 5.70, p-FWE = 0.0005), left rostral prefrontal cortex of the salience network (SN.RPFC L) (T(30) = 6.64, p-FWE = 0.0005), and left lateral parietal part of the DMN (LP.DMN L) (T(30) = 3.75, p-FWE = 0.12). This cluster also included negative connectivity between DMN.MPFC and the anterior cingulate cortex of the salience network (SN.ACC) (T(30) = −8.64, p-FWE < 0.001). The second cluster (mass = 126.78, size = 2, p-FWE = 0.001) included two positive connections between the posterior parietal cortex of the frontoparietal network, right inferior parietal sulcus of the dorsal attention network (T(30) = 7.59, p-FWE < 0.001), and right visual lateral network (T(30) = 8.31, p-FWE < 0.001) (Figure 4).

To explore the nature of the connectivity changes, we systematically varied a cluster-forming threshold with a step of 0.01 (10%). The NBS showed that applying a lower threshold (*p* < 0.05) supported the identification of this component. Moreover, applying a more conservative threshold (*p* < 0.0001) also did not change dramatically the structure of these clusters (Appendix A). These results indicated that the effect of cold-water immersion is associated with a strong and topologically consistent component of the interconnected clusters of networks.

### 3.4. Functional Connections Explaining Enhanced Positive Affect after CWI

We next tested associations between NBS connectivity measures [post-CWI > pre-CWI] and changes in positive affect (defined as differences in positive affect scores before and after cold-water immersion). The connectivity changes that linearly relate to changes in positive affect would indicate an association between them.

Thresholding the connectivity maps at *p* < 0.001 revealed two clusters. One cluster (mass = 102.86, size = 2, p-FWE = 0.006) included a positive connection between the medial prefrontal cortex of the default mode network (DMN.MPFC) and the left rostral prefrontal part of the salience network (SN.RPFC) (T(29) = 5.19, p-FWE = 0.003), and a negative connection between the DMN.MPFC and the anterior cingulate cortex of the salience network (SN.ACC) (T(29) = −8.35, p-FWE < 0.001). Another cluster (mass = 96.62, size = 2, p-FWE = 0.005) consisted of two positive connections between the posterior parietal part of the frontoparietal network (FP.PPC) and visual lateral network (T(29) = 7.89, p-FWE < 0.001), and the intraparietal sulcus of the dorsal attention network (T(29) = 6.37, p-FWE = 0.0002) in the right hemisphere (Figure 5).

The relationship between the changes in functional connectivity and the changes in positive affect after cold-water immersion was consistent across the sample (Figure 6).

We systematically varied a connection threshold with a 0.01 step. The results indicated that topologically focal connections formed the component because we found no evidence for connections at lower thresholds (Appendix A).

### 3.5. Functional Connections Explaining Reducing Negative Affect after CWI

Contrast [post-CWI > pre-CWI] with added scores of changes in negative affect as a covariate using a height threshold *p* < 0.001 revealed a topological cluster comprising one connection between DMN.MPFC and DMN.PCC (mass = 154.50, size = 1, p-FWE = 0.001) (Figure 7A). However, the effect size for this connection was very small and cannot be reliably estimated at the population level with 90% of confidence (Figure 8).

Systematically varying the connectivity threshold indicated that this effect occurred at a more conservative threshold (*p* < 0.001–0.0001). However, lowing the threshold showed a large and spatially extended component comprising 68 connections (mass = 594.50, p-FWE = 0.004) (Figure 7B) (Appendix A). As this component was presented across a range of thresholds, it was likely to be characterised by both subtle yet topologically extended differences and strong but topologically focal differences.

We further tested whether the connection between DMN.MPFC and DMN.PPC was specific to a reduction in negative affect by performing an additional NBS analysis, but this time controlling for changes in positive affect (contrast [Allsubjects (0) changes in positive affect (−1) changes in negative affect (1)] in [post-CWI > pre-CWI]). The results showed that after controlling for the positive affect, the component DMN.MPFC-DMN.PPC could no longer be identified. In contrast, controlling for the negative affect did not change the structure of a component associated with the increased positive affect.

## 4. Discussion

Here, we presented the results of the first study investigating the effects of short-term head-out whole-body cold-water immersion on functional connectivity for positive and negative affect in healthy adults unhabituated to cold water.

Our recordings of the physiological responses to cold water showed changes in cardiorespiratory function, indicating that all participants experienced a cold shock. This finding aligns with earlier research reporting similar cold-water shock responses such as inspiratory gasp, hyperventilation, and increasing heart rate [2,49]. It has to be noted that although the magnitude of cold shock varied considerably from participant to participant, the overall physiological changes observed in the present study (~20 °C) were compatible with studies using a lower water temperature (~10 °C) [50]. What is unclear is whether humoral control mechanisms and the activity in the sympathetic nervous system with higher and lower water temperatures are the same or associated with differential biochemical changes. Examining the relationship between water temperature, physiological responses, and biochemical changes in future studies may ensure a better understanding of the effects of cold-water immersion on brain functioning.

The results of the self-report mood questionnaire showed a significant increase in positive affect and a decrease in negative affect after cold-water immersion. This finding is well-aligned with two previous studies that reported acute positive changes in mood following a single immersion in cold water [12,13]. Our data suggest that the change in positive affect was mainly associated with increasing alertness, and feeling more inspired, active, attentive, and proud, whilst the decreasing negative affect reflected feeling less nervous and distressed. Notably, the positive changes in affective state after cold-water immersion tap mood states that are typically reduced in depressive disorders. For example, reduced energy level, motivation, and alertness, and elevated emotional disturbance are well-recognised symptoms of mental health conditions such as major depression [51]. A recent study investigating the perceived effects of regular exposure to cold water on health and well-being in a large population reported a significant reduction in depressive symptoms in 59% of participants [52]. Our finding indicates that cold-water immersion may target specific mood states, providing more explanations for the widely reported enhanced mood following short-term immersion in cold water, and may have a therapeutic effect in some. It has to be noted that no participants in the present study showed the opposite effect of cold-water bath (i.e., decreasing in positive affect while increasing in negative affect). However, developing our knowledge about how exposure to cold water affects the emotional state will be incomplete without testing the opposite effect.

It is worth mentioning that the ‘broaden and build’ model of positive emotions argued that the ratio of positive to negative affect could distinguish well-functioning individuals from others and proposed a critical ratio of 3:1 [53]. Our data show changes in the ratio from 1.75 (29.28:16.75) in the pre-CWI assessment to 3.00 (36.28:12.09) in the post-CWI assessment, indicating the perceived positive effects of cold-water immersion. The scores for the pre-CWI positive and negative affect closely resemble PANAS scores in the previous study (i.e., 34.04 for positive affect and 18.93 for negative affect, ratio = 1.79) investigating affect and the brain’s functional organisation in a group of people within the same age range and gender ratio as in the present study [35]. Important, we did not find differences in the positive and negative affect between participants with previous experience in neuroimaging procedures and naïve to the scanning session. Therefore, it is unlikely that the pre-CWI PANAS scores reflected negative emotions in anticipation of the scanning procedures (e.g., anxiety, scanning environment).

Consistent with the bivalence hypothesis, we found that positive and negative affective experiences were uncorrelated in both (pre-CWI and post-CWI) assessments. In contrast to previous studies reporting a negative correlation between the positive and negative affect [54], we found no evidence for the association. Importantly, we also found no relationship between changes in the positive and negative affect states, suggesting the independence of these two constructs.

Our NBS results provide evidence that exposure to cold water was associated with a temporal correlation across multiple neural networks, including the medial prefrontal cortex of the default mode network, left anterior insula, left rostral prefrontal cortex and anterior cingulate cortex of the salience network, and left lateral parietal part of the DMN. However, only four connections were uniquely correlated with self-reported changes in positive affect after cold-water immersion: between DMN.MPFC; two nodes of the salience network (ACC and RPFC-L); and between FP.PPC, DAN.IPS, and Vis.Lat in the right hemisphere. This finding is in line with the broad literature on the organizational principles of the human brain’s functional connectome during the processing of affective information [24,55,56]. For example, several studies have shown that top-down attention to one’s own emotional state amplifies activity within the MPFC part of the DMN and regions such as the ACC, PPC, and occipital cortex [57]. The coupling between the RPFC, MPFC, and posterior parietal cortices during emotion processing has been well-established in healthy individuals, whilst aberrant functional connectivity between these nodes was linked to various depressive symptoms [58]. It has to be noted that research collected considerable evidence of a positive coupling between the MPFC and ACC for processing negative emotional information [59]. It was proposed that the interaction between them is implicated in regulatory functions such as regulating negative affect during social rejection [60], and the extinction of arousal caused by emotional stimuli [61]. Our finding of the reverse coupling between the MPFC and ACC associated with an increase in positive affect suggests that positive affect requires less regulation, and the direction of the relationship may be adjusted flexibly depending on the affective valence. If replicated, the finding that cold-water immersion triggers the reversed coupling between the MPFC and ACC via the increased positive affect may be of clinical interest to facilitate cognitive flexibility and reduce depressive rumination [62].

Functional connections between the posterior parietal cortex of the FP network and the inferior parietal sulcus of DAN have long been recognised as a neural substrate for alerting, orienting, and executive control of attention [63]. These findings mainly came from studies using various goal-directed tasks required to engage attentional control. Here, we observed a strong coupling between these nodes without any explicit task. Moreover, during resting-state scans, participants often experience difficulty maintaining a constant level of wakefulness. For example, a study using 1147 datasets collected during a resting state revealed a loss of wakefulness in a third of subjects within 3 min, and half of the participants lost wakefulness after 10 min of scanning [64]. Our finding of increased connectivity between FP.PCC and DAN.IPS during a 13 min resting-state scan can be attributed to the arousing effect of cold water, which is well-aligned with participants’ report of feeling more alert, active, and attentive after the CWI procedure. The physiological responses to cold-water immersion have been extensively studied and are known to have arousing effects resulting in peripheral vasoconstriction, tachycardia, hyperventilation, hypertension [2], and spikes in norepinephrine and dopamine concentrations [40] upon initial immersion; collectively, these responses are termed the cold shock response.

The connection between the FP.PCC and the visual lateral network likely reflect our resting state with open eyes. The functional connectivity between attention and the visual system has been consistently reported in open-eyes compared to closed-eyes resting-state scans [38].

One interesting observation is that these four functional connections formed a component of interacting networks with a strong focal effect. Although the specific characteristics of focal vs. distributed connectivity between interacting networks and their relation to behavioural effects remain largely unknown, our two findings support this observation. First, the effect sizes of individual connections indicated that changes in positive affect could reliably predict the connectivity strength within the component, yielding a medium effect across all connections. Second, the relationship between the changes in positive affect and functional connectivity within the component was consistent across the sample. Notably, 90.32% of participants showed connectivity between the DMN.MPFC and SN.ACC and 80.64% showed connectivity between the DMN.MPFC and SN.RPFC-R. The coupling between FP.PCC-R, DAN.IPS-R, and Vis.Lat-R was found in 83.87% of participants. In terms of biological information processing, focal vs. distributed connections are close to the concept of entropy as a measure of complexity, randomness, or predictability of a dynamic process [65]. The considerable effect size and the consistency of the relationship between the changes in positive affect and the component of interacting networks after cold-water immersion may indicate certainty and predictability of the effect (focal connectivity effect, lower entropy). In contrast, weak but more complex connections can be associated with uncertainty and lower predictability (distributed connectivity effect, higher entropy). Here, we use the analogy with entropy to explain the difference between focal vs. distributed connectivity. However, the plausibility of this analogy is supported by studies in animals and human subjects showing a negative correlation between the strength of functional connectivity and regional entropy as a measure of spatiotemporal variability [66].

Our data do not provide evidence of unique functional connectivity for changes in the negative affect. The small component comprising the MPFC and PCC of the DMN could not be identified after controlling for changes in the positive affect. Lowering the threshold by means of detecting components formed by subtle interactions between networks showed a large component of distributed connections. Although identifying these connections may not be clinically meaningful, examining the property of this inherently ‘fuzzy’ connectivity may be of potential interest. For example, the complexity of the low-level interactions may be functionally relevant to determining the onset of mood disorders or identifying the ‘lower layer’ of negative affective processing. It has to be noted that this is not the first observation of complex low-level functional connectivity for negative affect. For example, Tozzi and colleagues [67] reported the low canonical variate loading of multiple brain regions at rest for negative emotions. A widely distributed component co-varied with a behavioural bias for sad vs. neutral emotion was also reported in a study examining self-prioritisation effects [55].

Limitations and Directions for Future Research:

The neuroimaging literature is highly inconsistent in the precision mapping of brain networks. The substantial disparity in parcellation scales and nomenclature across existing brain atlases [68] limited comparisons between our study and previous work in affective processing. The analytical approach in our study focused on large-scale cortical networks to examine the effects of cold-water immersion on functional connectivity for positive and negative mood changes. Although this approach allowed us to pin down these effects in the cerebral cortex, defining the contribution of subcortical networks is challenging. There are several reasons for this. The work on mapping the subcortical connectivity to well-established large-scale networks has just begun. For example, recent work using high-resolution functional imaging mapped the subcortical connectivity of the default mode network [69]. However, the subcortical connectivity to other large-scale networks remains largely unknown. Moreover, the BOLD signal varies considerably between cortical and subcortical areas in noise and amplitude [70]. For example, studies reported the low reliability of functional connectivity for subcortical connections compared to cortical connections [71,72]. The low reliability was linked to factors such as unique activity, proximity to non-neuronal sources (e.g., susceptibility variations associated with breathing, cerebrospinal fluid), or a lower central signal-to-noise ratio with highly parallel array coils [70]. Although these factors have been discussed to improve mapping cortical and subcortical networks (68), this work is still in progress.

The present study is not a randomised control study, as a control group was not included in the design for three reasons. First, previous studies demonstrated a stable pattern of network connections between two consecutive scan sessions on the same day, and a scanning session occurred in 2–3 months [73]. A recent study using large sample longitudinal data provided striking evidence that resting-state functional connectome is stable over months to years [74]. These findings indicate the stability of the individual functional architecture of large-scale networks. Although individual variations may occur from measurement to measurement, these variations are negligible compared to those triggered by an external event, such as a cold-water bath in the present study, affecting physical, emotional, and cognitive states leading to meaningful changes in the network architecture. Second, defining a meaningful procedure inducing physiological stress to the body while allowing positive and negative mood changes is challenging. Third, we focused on how changes in self-reported positive and negative affect triggered by physiological stress were linked to changes in functional connectivity between large-scale brain networks. Although the self-reported positive changes in mood in the present study are compatible with those reported in studies of open-water immersion, it remains unknown whether social or psychological factors might have confounding effects. For example, the courage to take the challenge or other intrinsic motivations might facilitate the positive affect. Therefore, providing a direct link between the physiological effects of cold-water immersion and changes in brain connectivity still requires systematic investigations. Moreover, a replication study is necessary to confirm our results.

## 5. Conclusions

Our findings suggest that short-term head-out immersion in cold water is associated with a facilitated positive affect and reduced negative affect. These changes in affective states are linked to changes in connectivity between large-scale brain networks. Positive affect is supported by an interaction between the ‘core’ networks involved in affective processes such as the default mode, salience, frontoparietal, and visual lateral networks. Changes in negative affect are associated with a distributed component of interacting networks at a reduced threshold. The changes in positive and negative affect after cold-water immersion occur independently of each other, supporting the bivalence model of affective processing. These findings contribute to our understanding of the effects of whole-body cold-water immersion on brain functioning.

## Figures and Tables

**Figure 1 biology-12-00211-f001:**
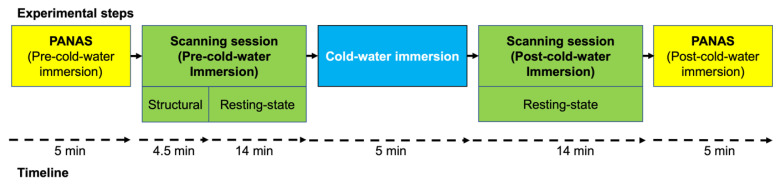
Experimental steps and timeline.

**Figure 2 biology-12-00211-f002:**
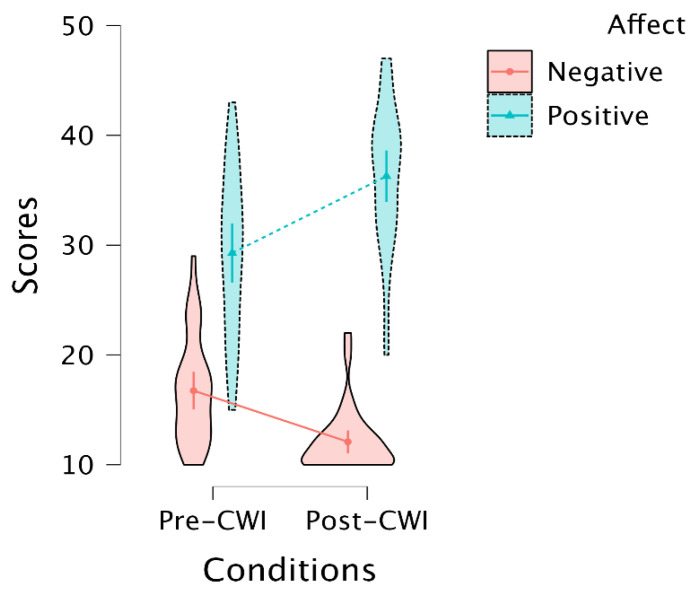
Violin plot depicting the PANAS scores of positive and negative affect during post- and pre-cold-water immersion (see details in Appendix A). Dots inside the violins represent the means of the estimates with 95% confidence intervals.

**Figure 3 biology-12-00211-f003:**
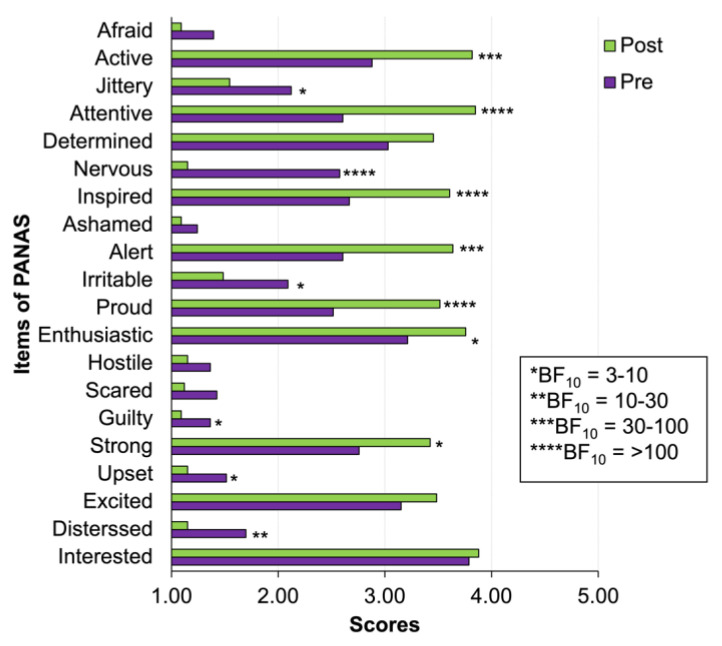
Clustered bar plot of individual items of PANAS (see details of Wilcoxon’s signed rank statistics and Bayes factor in Appendix A). Stars indicate the value of the Bayes factor for the alternative hypothesis of the difference between pre-CWI and post-CWI scores.

**Figure 4 biology-12-00211-f004:**
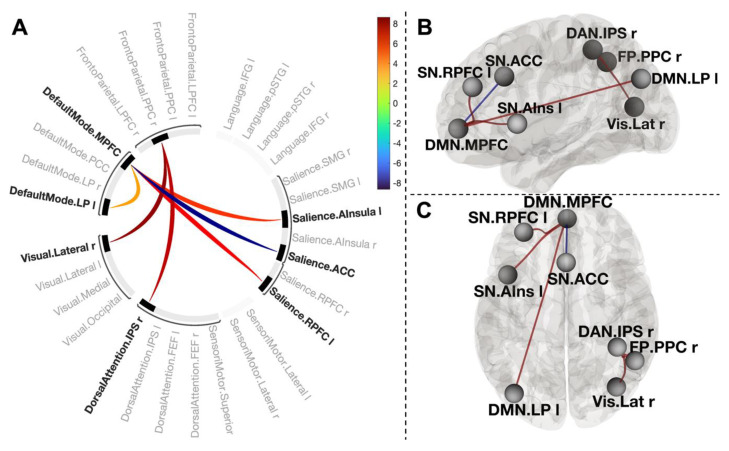
Connectogram of networks (**A**) and corresponding glass brain (grey matter, high-resolution mask, (**B**)—sagittal view, (**C**)—axial view) for contrast [post-CWI > pre-CWI]. Vertical colour bars indicate T-test statistics for individual connections. Glass brain figures visualise the spatial location of connections comprising each component, where a sphere represents the centre of the corresponding network. The red and blue lines depict positive or negative correlations between networks.

**Figure 5 biology-12-00211-f005:**
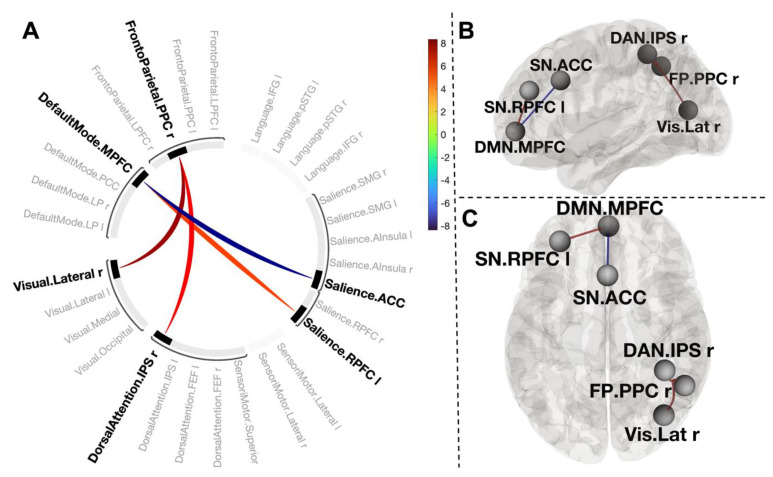
Connectogram of networks (**A**) and corresponding glass brain (grey matter, high-resolution mask, (**B**)—sagittal view, (**C**)—axial view) for contrast [post-CWI > pre-CWI] with changes in self-report of positive affect scores as a covariate. Vertical colour bars indicate T-test statistics for individual connections. Glass brain figures visualise the spatial location of connections comprising each component where a sphere represents the centre of the corresponding network. The red and blue lines depict positive or negative correlations between networks.

**Figure 6 biology-12-00211-f006:**
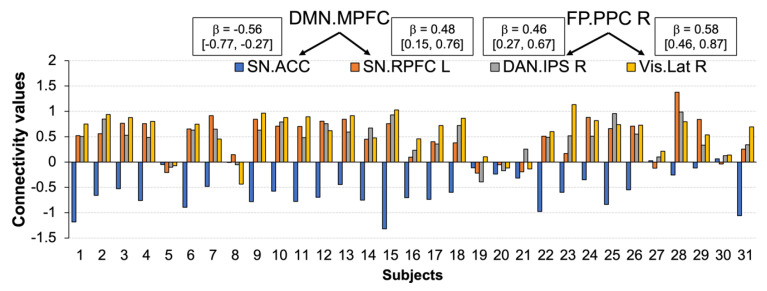
Individual connectivity values for changes in positive affect after cold-water immersion. The connectivity values (Y-axis) represent the Fisher-transformed correlation coefficients. Rectangles depict effect size with 90% CI for the corresponding connection.

**Figure 7 biology-12-00211-f007:**
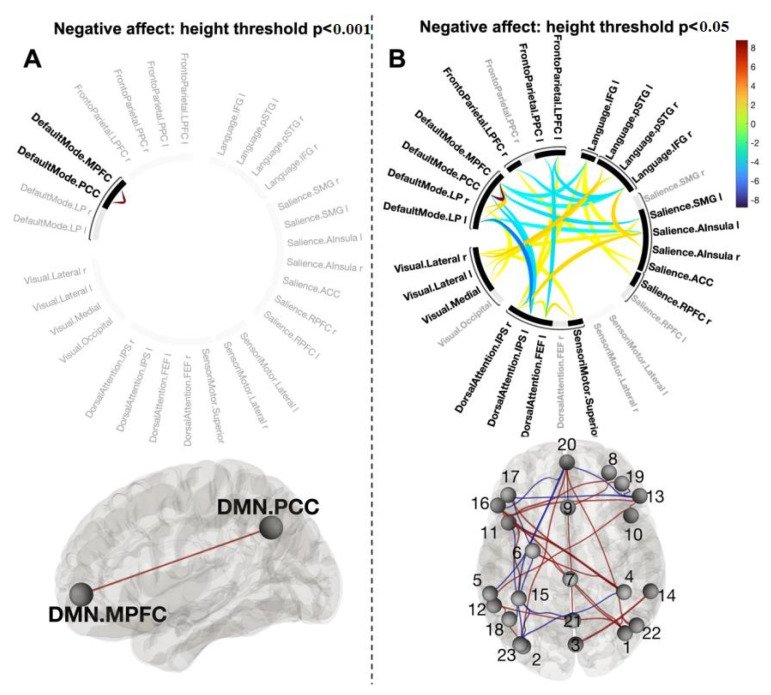
Connectogram of networks and corresponding glass brain (grey matter, high-resolution mask) for contrast [post-CWI > pre-CWI] with changes in self-report of negative affect scores as a covariate for height threshold *p* < 0.001 (**A**) and *p* < 0.05 (**B**). Vertical colour bars indicate T-test statistics for individual connections. Glass brain figures visualise spatial location of connections comprising each component where a sphere represents the centre of the corresponding network. The red and blue lines depict positive or negative correlations between networks. 1—Vis. Lat (R), 2—Vis. Lat (L), 3—Vis. Med, 4—DAN.IPS (R), 5—DAN.IPS (L), 6—DAN.FEF (L), 7—SM.Superior, 8—SN.RPFC (R), 9—SN.ACC, 10—SN.AIns (R), 11—SN.AIns (L), 12—SN.SMG (L), 13—Language.IFG (R), 14—Language.pSTS (R), 15—Language.pSTS (L), 16—Language.IFG (L), 17—FP.LPFC (R), 18—FP.PPC (L), 19—FP.LPFC (L), 20—DMN.MPFC, 21—DMN.PCC, 22—DMN.LP (R), 23—DMN.LP (L).

**Figure 8 biology-12-00211-f008:**
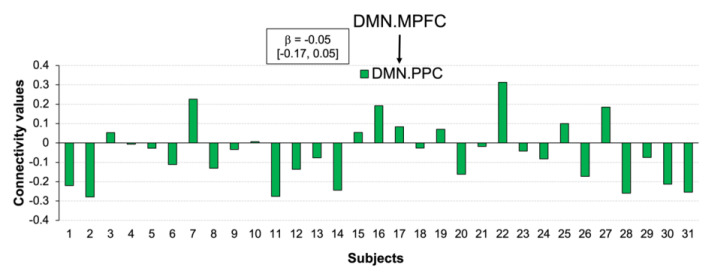
Individual connectivity values for changes in negative affect after cold-water immersion. The connectivity values (Y-axis) represent the Fisher-transformed correlation coefficients. Rectangle depicts effect size with 90% CI for the corresponding connection.

## Data Availability

Available on reasonable request to the corresponding author. A detailed protocol of the study will be published shortly in Bio-Protocols.

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
