# Peer review of "Short-Term Head-Out Whole-Body Cold-Water Immersion Facilitates Positive Affect and Increases Interaction between Large-Scale Brain Networks"

_biology, 2023, doi:10.3390/biology12020211_

Round 1
Reviewer 1 Report
This manuscript studied the alteration of functional connectivity after short-term cold water immersion using resting state fMRI. The results demonstrate that several clusters of functional connectivity change before and after 5 min cold water immersion. This is a well-design and interesting study. It is well presented and I have several concerns:
Major:
1. The author chose the CONN network cortical ROIs to contruct the functional networks. The CONN network atlas only contain coritcal regions without the subcortical regions. The atlas also only covers part of the cortical regions rather than the whole cortical surface. Since the subcortical regions play a vital role in many emotinal regulations and physiological responses to the environment. The authors' introduction also present that other similar study found that functional activities in subcortical regions altered. So the lack of subcortical regions would be a problem and lost important information that would related to the goal of this study. The author may want to repeat or valid the current results with another brain atlas covered the subcortical regions.
Minor comments:
1. In the fMRI data processing part, the author chose a threshold of 0.9 mm for the FD instead of the commonly used 0.5 mm. Could you explain the reason for that? How many percentages of volumes exceeded the 0.5 mm threshold?
2. In the same part as NO.1 comments. "This procedure was applied separately to the functional data, using the mean BOLD signal as a reference image, and to the anatomical data, using the T1-weighted volume as a reference image. " It's hard to understand what's the processing here, please reword it.
3. The author choose the CONN network cortical regions
3. Network analysis part, "In total, we analysed 830 connections across 30 ROIs." the number of connections is wrong, it should be N*(N-1)/2, please correct it.
4. In supplementray materials, Box4, Box.5, since the conection here are not directed or effective connection, "-> " sign may confuse the readers.
5. The figures show changes of FC, the author could use the width of connections to represent the Tvalue of each connection, that would make the figures more inforamative.
6. MRI acquistions: matrix size = 64 × 64 mm2; the matrix size should not have unit. "spacing between slices 2.5 mm", please clarify that do it mean the gap between slices is 2.5mm. Please add the FOV or inplane resolution for fMRI
Author Response
We want to thank Reviewer 1 for their time, engagement, and constructive comments on our work. We have carefully addressed all issues point-by-point (please, see attached file), where Reviewers’ comments are highlighted in bold. In the revised version of the manuscript, the changes we have made are highlighted in red.
We hope that you and the Editor will find these changes satisfactory and that the paper is now acceptable for publication.
We look forward to hearing from you.
Yours faithfully,
Ala Yankouskaya, Ruth Williamson, Cameron Stacey, John Totman and Heather Massey

Reviewer 2 Report
Functional magnetic resonance imaging (fMRI) was used to identify the topological clusters of networks combined with behavioral changes in positive and negative affect after a 5-minute cold water immersion. The results of the self-report mood questionnaire showed a significant increase in positive affect and a decrease in negative affect after cold water immersion. The increase in positive affect was supported by a unique component of interacting networks, including the medial prefrontal node of the DMN, a posterior parietal node of the fronto-parietal network, anterior cingulate and rostral prefrontal parts of the Salience Network and Visual Lateral Network. Changes in negative affect were associated with a distributed component of interacting networks at a reduced threshold. Affective changes after cold water immersion occurred independently, supporting the bivalence model of affective processing.
I got a good impression of reading the manuscript. The topic is really interesting for understanding the brain mechanisms of cold impact on the personal emotional state. The experimental model, data collection and analysis methods and results are well described. All statistical comparisons are mathematically correct. The results are presented in the figures so that they are easy for a reader to understand.
I have two minor remarks to improve the manuscript.
First, as the authors noted, the response to cold water immersion can vary greatly between subjects. For most participants, cold water immersion caused an increase in positive affect and a decrease in negative affect. However, it is possible that there were people with the opposite impact of cold water immersion, i.e. they could have an increase in negative and a decrease in positive affect. Were such participants detected in the experimental group of this study? If so, is the change in brain connectivity in such participants different from the pattern found in most participants? I recommend that the authors mention this issue in the discussion section.
Secondly, the authors did not conduct a control experiment with re-registration of fMRI without immersion in cold water. Hypothetically, changes in brain connectivity on fMRI could result from the subject's re-participation in the scan. I recommend finding the references in the literature indicating the stability of the connectivity pattern of different brain networks on repeated fMRI recordings and leading these findings into discussion. This will justify that a follow-up experiment with repeated fMRI recording without cold water immersion for this study is unnecessary.
Author Response
We want to thank Reviewer 2 for their time, engagement, and constructive comments on our work. We have carefully addressed all issues point-by-point (please, see attached file), where Reviewers’ comments are highlighted in bold. In the revised version of the manuscript, the changes we have made are highlighted in red.
We hope that you and the Editor will find these changes satisfactory and that the paper is now acceptable for publication.
We look forward to hearing from you.
Yours faithfully,
Ala Yankouskaya, Ruth Williamson, Cameron Stacey, John Totman and Heather Massey

Round 2
Reviewer 1 Report
The authors have addressed my concern.
Author Response
Thank you for spotting the issues listed above.
1. We corrected ‘exceeding 9 mm’ to ‘exceeding 0.9 mm’.
2. We adjusted the format of Figure 6.
3. Original: 'Although these and other issues relating to mapping cortical and subcortical networks have been widely discussed (65), we are still far from developing a full account.'
Corrected: Although these factors have been discussed to improve mapping cortical and subcortical networks (65), this work is still in progress.
4. Original sentence ..‘the luck of previous studies’ ‘limited literature’ difficult to relate’, ‘the field’ has been removed.